# GCN sensitive protein translation in yeast

William A. Barr[1¤a], Ruchi B. Sheth[1☉], Jack Kwon[1☉], Jungwoo Cho[1], Jacob W. Glickman[1¤b], Felix Hart[1], Om K. Chatterji[1], Kristen Scopino[1], Karen Voelkel-Meiman[2], Daniel Krizanc[3,4], Kelly M. Thayer[3,4,5], Michael P. Weir[1,4]*

**1** Department of Biology, Wesleyan University, Middletown, CT, United States of America, **2** Department of Molecular Biology and Biochemistry, Wesleyan University, Middletown, CT, United States of America, **3** Department of Mathematics and Computer Science, Wesleyan University, Middletown, CT, United States of America, **4** Department of Chemistry, Wesleyan University, Middletown, CT, United States of America, **5** College of Integrative Sciences, Wesleyan University, Middletown, CT, United States of America

☉ These authors contributed equally to this work.
¤a Current address: Laboratory of Molecular Metabolism, The Rockefeller University, New York, NY, United States of America
¤b Current address: Icahn School of Medicine at Mount Sinai, New York, NY, United States of America
* mweir@wesleyan.edu

**Data Availability Statement:** All relevant data are within the paper and its Supporting Information files.

**Funding:** MPW GM120719 National Institutes of Health https://www.nih.gov KMT GM128102

## Abstract

Levels of protein translation by ribosomes are governed both by features of the translation machinery as well as sequence properties of the mRNAs themselves. We focus here on a striking three-nucleotide periodicity, characterized by overrepresentation of GCN codons and underrepresentation of G at the second position of codons, that is observed in Open Reading Frames (ORFs) of mRNAs. Our examination of mRNA sequences in *Saccharomyces cerevisiae* revealed that this periodicity is particularly pronounced in the initial codons—the ramp region—of ORFs of genes with high protein expression. It is also found in mRNA sequences immediately following non-standard AUG start sites, located upstream or downstream of the standard annotated start sites of genes. To explore the possible influences of the ramp GCN periodicity on translation efficiency, we tested edited ramps with accentuated or depressed periodicity in two test genes, *SKN7* and *HMT1*. Greater conformance to $(GCN)_n$ was found to significantly depress translation, whereas disrupting conformance had neutral or positive effects on translation. Our recent Molecular Dynamics analysis of a subsystem of translocating ribosomes in yeast revealed an interaction surface that H-bonds to the +1 codon that is about to enter the ribosome decoding center A site. The surface, comprised of 16S/18S rRNA C1054 and A1196 (*E. coli* numbering) and R146 of ribosomal protein Rps3, preferentially interacts with GCN codons, and we hypothesize that modulation of this mRNA-ribosome interaction may underlie GCN-mediated regulation of protein translation. Integration of our expression studies with large-scale reporter studies of ramp sequence variants suggests a model in which the C1054-A1196-R146 (CAR) interaction surface can act as both an accelerator and braking system for ribosome translation.

National Institutes of Health https://www.nih.gov The funders had no role in study design, data collection and analysis, decision to publish, or preparation of the manuscript.

**Competing interests:** The authors have declared that no competing interests exist.

## Introduction

Most gene expression requires ribosome translation of mRNA sequences into polypeptides. Although many features of this process are very well described, there are mechanisms governing translation rate, particularly at the genomic scale, that remain to be discovered. Different mRNAs can vary 10–100 fold in their translation efficiency [1–3]. Translation efficiency is influenced by biophysical parameters and informational properties of the mRNA molecule. Low mRNA stability and strong secondary structure can have negative effects on translation efficiency that are largely independent of the sequence contained within the mRNA [4, 5]. Sequence determinants of translation efficiency are less well-described but are increasingly accepted as playing important roles in the translation mechanism in disparate taxa [6–9].

Advances in functional genomics have confirmed that factors such as codon usage, tRNA abundance, and 5' UTR sequence can significantly affect translation efficiency and explain some of the variation in translation efficiency across the transcriptome [10–12]. However, it is estimated that only 60% of the variation in translation efficiencies are explained by these known sequence determinants [3]. Furthermore, the descriptions of these factors generally do not address the influences of sequence elements within the Open Reading Frame (ORF) that span regions larger than individual codons. For example, dicodon frequencies can have strong influences on translation efficiency [13, 14]. In addition, the first few codons downstream of the translation initiation site—the ramp region—are thought to be translated at slower rates than the rest of the mRNA sequence [1] and to be more likely to form secondary structures [15].

Another broad-scale feature of mRNA sequences is the overrepresentation of in-frame $(GCN)_n$ motifs [16, 17]. While this striking feature of ORFs has been known for many years, its possible functional significance in protein translation has been unclear. Initial hypotheses focused on GCN periodicity as a framing code for the ribosome, but low-throughput assays failed to establish a functional relationship between GCN periodicity and reading frame maintenance [18]. In the 1990s, before ribosome structures were known, it was suggested that the GCN periodicity might relate in some way to a conserved motif in 16S and 18S ribosomes, the 530 loop (16S numbering). This loop, located in helix 18 of 16S/18S rRNA, has sequence complementarity to GCN [16, 19]. Moreover, cross-linking studies suggested that the 530 loop associates with mRNA codons just 3' of the A-site codon [20, 21]. With the elucidation of ribosome structures [22–26], the 530 loop was found to be located at the surface of the mRNA entrance tunnel adjacent to the decoding center A-site (Fig 1). However, crystal and cryo-EM ribosome structures have not shown base pairing between the 530 loop rRNA and mRNA.

Given these intriguing observations, we decided to examine the dynamic behavior of a subsystem of the ribosome centered around the A-site decoding center and 530 loop neighborhood using Molecular Dynamics (MD) simulations [27]. These MD experiments have shown that rRNA nucleotides in the 530 loop tend to stay base paired with each other in a standard configuration. However, in the course of these MD experiments, we discovered a new ribosome-mRNA interaction surface located between the A-site tRNA and the 530 loop (Fig 1). This interaction surface is made up of three stacked residues—C1054 and A1196 of 16S/18S rRNA (*E. coli* 16S numbering), and R146 of ribosomal protein S3 (yeast Rps3 numbering). C1054 and A1196 are conserved in prokaryotes and eukaryotes (A1196 is occasionally replaced with U1196, e.g. in *T. thermophilus*) and R146 is conserved in eukaryotes. At the beginning of translocation, the C1054-A1196-R146 (CAR) interaction surface—consisting of the Watson-Crick edge of C1054, the Hoogsteen edge of A1196, and an edge of the guanidinium group of R146— is anchored through base stacking to the wobble anticodon nucleotide (tRNA nucleotide 34), and H-bonds to the +1 codon about to enter the A site in a sequence-sensitive

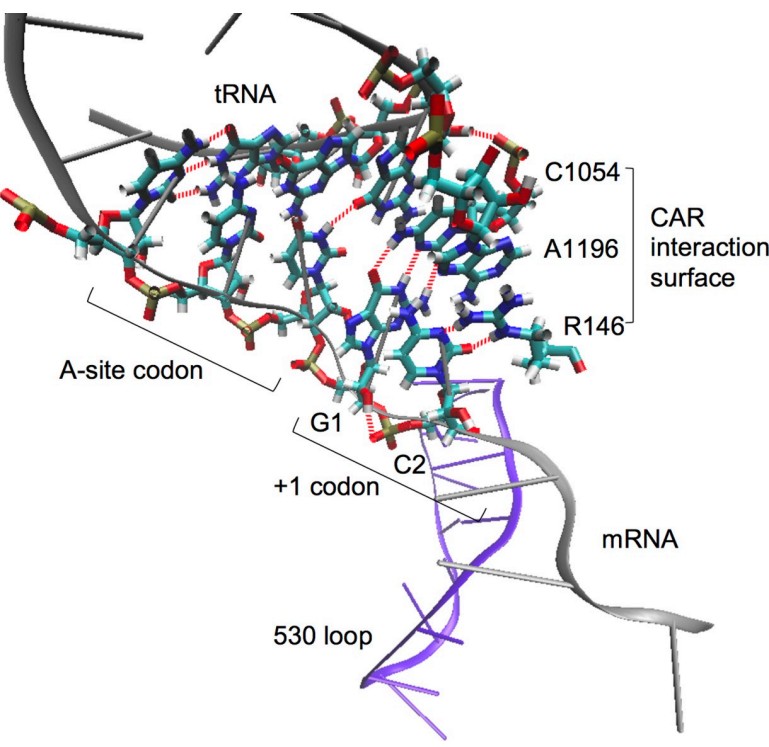

**Fig 1. The CAR interaction surface during ribosome translocation.** Molecular Dynamics analysis [27] of a subsystem of the translocating ribosome encompassing the A-site decoding center neighborhood showed that the +1 mRNA codon about to enter the A site H-bonds (red dashes) to the CAR interaction surface consisting of 16S/18S C1054, A1196 (*E. coli* numbering) and yeast ribosome protein Rps3 R146. Stacking interactions provide structural integrity to the CAR surface and anchor it to the wobble position anticodon nucleotide (tRNA nucleotide 34). The CAR surface preferentially H-bonds to GCN codons [27]. The 530 loop (violet) lies adjacent to the mRNA. The structure is viewed with Visual Molecular Dynamics (VMD; [28, 29]).

interaction (Fig 1). The MD experiments show that if GCN at the +1 codon is replaced with GGN, GAN or GUN, the CAR-mRNA interaction is much weaker [27].

With the discovery of the CAR surface, we decided to explore in this study the potential functional significance of GCN periodicity in translation. We found that the GCN periodicity is enhanced in the initial codons of ORFs—the "ramp" region [30, 31]—and this trend is particularly pronounced in genes with higher protein expression. We also observed that GCN periodicity is enhanced after non-standard translation start codons, defining ORF starts that are either upstream or downstream of the standard annotated start codon. By changing the GCN periodicity in the ramp regions of two test genes—*SKN7* and *HMT1*— we found that translation is depressed in ramp mutants with more pronounced GCN, and translation is enhanced when G and C frequencies are reduced (at positions 1 and 2 respectively) in ramp codons. These observations are integrated with large-scale reporter studies of ramp sequence variants [13, 31], suggesting a model in which the CAR interface can act both as an accelerator and braking system for ribosome elongation.

## Results

### GCN periodicity is pronounced in the 5' ramp region of highly-expressed ORFs

Annotated ORFs in the yeast genome, aligned at their AUG start or stop codons (Fig 2A), show a 3-nucleotide GCN periodicity in which G at position 1 (G1) and C at position 2 (C2) of

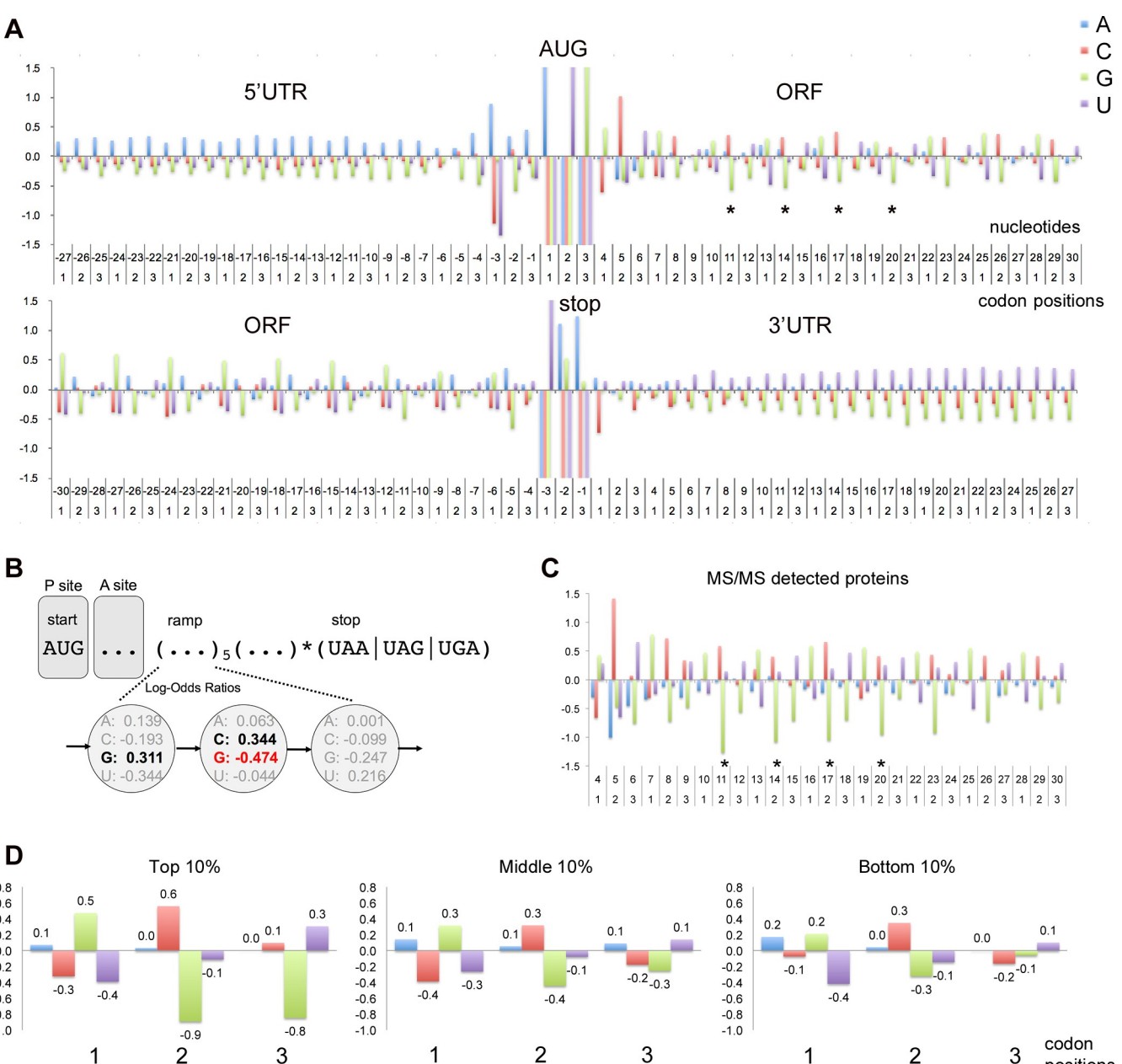

**Fig 2. GCN periodicity in ORFs.** (A) Three-nucleotide periodicity spans ORFs but not UTRs. ORFs of 5791 yeast genes were aligned at their AUG start codon (positions 1, 2, 3 in first line) or their stop codon (positions -3, -2, -1 in second line). The log-odds ratio graphs of $\log_2(\text{freq}_{obs}/\text{freq}_{background})$ show at each nucleotide position the deviations from the background nucleotide frequencies in ORFs (A: 0.326; C: 0.192; G: 0.204; U: 0.278). A 3-nucleotide periodicity is detected throughout the ORFs characterized by depression of G at the second nucleotide of codons (*) and enhancement of G and C at positions 1 and 2 respectively. The periodicity is not seen in the 5'UTR or 3'UTR. The yeast Kozak sequence surrounding the start codon (AAA(A/G)AA<u>AUG</u>CG) has pronounced depression of C/U at position -3. (B) A cartoon depiction of an mRNA at the beginning of translation. Initial codons in the ramp region show log-odds ratios illustrating a GCN periodicity and depression of G at position 2. (C) Amino termini of 1027 annotated yeast proteins were detected by peptide mass spectrometry [32]. ORFs aligned at their start codons (nucleotides 1, 2, 3) are displayed to show log-odds ratio deviations from the background nucleotide frequencies in ORFs. Pronounced depression of G at position 2 of codons is observed between nucleotides 10 and 20 (*) when compared to the full set of 5791 verified and uncharacterized yeast genes (in panel A). (D) The 3-nucleotide periodicity is pronounced in genes with high protein expression. Protein expression levels were determined previously in genomic-scale Western analysis of TAP-epitope tagged ORFs [33]. Periodicity was examined for 3838 verified and uncharacterized genes with detectable protein expression (protein expression levels > 0). These genes were partitioned into 10% bins, and superimposed codons between nucleotides 7 and 22 for all genes in each bin are illustrated. The periodicity increases progressively in strength as illustrated by the G2 depression. Bootstrap analysis indicates that compared to all genes, the average G2 depression between positions 7 and 22 is significantly enhanced in the MS/MS detected genes (C; *P* < 0.001) and top 10% protein expression bin (D; *P* < 0.001).

codons are overrepresented, and G at position 2 (G2) is underrepresented, compared to the background nucleotide frequencies in ORFs. The periodicity is notably absent from 5' and 3' UTRs (Fig 2A). The GCN periodicity was particularly pronounced in the initial codons, the ramp region, of ORFs (Fig 2B). Indeed, the periodicity, measured by Log-Odds Ratios (LOR), was significantly elevated (Fig 2C) in the ramp ORF regions of genes whose proteins were detected in cell lysates by peptide MS/MS spectrometry [32], suggesting that the periodicity is more pronounced in genes with high protein expression (and hence more readily detectable by MS/MS). This conclusion was confirmed by examining the ramp ORF periodicities of genes known to have high, intermediate and low protein expression (Fig 2D), based on expression levels reported in genome-wide Western analysis of 3,838 epitope-tagged yeast genes [33]. LOR scores for superimposed codons 3–7 (Fig 2D) showed that the top 10% of protein expressers had significant overrepresentation of G1 and C2, and underrepresentation of G2. The depression of G at position 2 was determined to be statistically significant (p<0.001) by bootstrap analysis. The top 10% also had significant depression of G at position 3 of ramp codons.

## ORFs of non-standard translation start sites have GCN periodicity

GCN periodicity is also detected in coding regions after non-standard translation initiation sites located upstream or downstream of the normal annotated start sites (Fig 3A; [32, 34]). Upstream ORFs (uORF) have been shown to play roles in modulating protein expression of the main protein products of various genes [35]. Our analysis of yeast MS/MS data revealed peptides that map to translation initiation sites of upstream ORFs (S1 and S2 Files) and these uORFs show pronounced G2 depression after their start codon (Fig 3B). This provides striking contrast to the general absence of 3-nucleotide periodicity in 5'UTRs (Fig 2A) and the significantly less pronounced periodicity observed when all upstream ORFs in known 5'UTRs are examined (Fig 3B).

Translation start codons located downstream of the annotated start codon can give rise to truncated proteins, when in the same reading frame, or novel protein products, including miniproteins [38], when in another reading frame. Downstream initiation can occur when scanning ribosomes skip the annotated start codon—leaky scanning [35]. When the downstream AUG is in the same reading frame as the main protein (frame 1), the 3-nucleotide periodicity is more pronounced than observed for the full set of frame-1 downstream AUGs screened in the MS/MS experiment (Fig 3C). When the downstream start codon is in a different reading frame, there is a pronounced depression of G nucleotides at the second position of codons starting 8 nucleotides after the start codon (Fig 3D). Although the start codon is in a different reading frame, the G2 depression is in the new reading frame, not the underlying frame-1 reading frame. Randomly-selected frame-2 AUGs from the ORFs of proteins do not show this G2 depression (Fig 3D). This result has been demonstrated for both *Saccharomyces cerevisiae* (Fig 3D) and *Drosophila* [32].

## Ramp GCN mutations change protein translation levels

Given these striking observations of GCN periodicities in the ramp regions of ORFs, we decided to assess the functional significance of the periodicity by mutating the ramp sequences in two test genes, *SKN7* which normally exhibits modest levels of protein expression, and *HMT1* which normally is expressed at high levels (S1 Fig). Both genes were tested in their normal chromosome environments, and both had C-terminal TAP epitope tags [33] to facilitate protein detection. Semi-quantitative Western blot analysis was performed to assess protein expression of ramp mutants relative to wildtype (WT) strains, using expression of Tub1p in

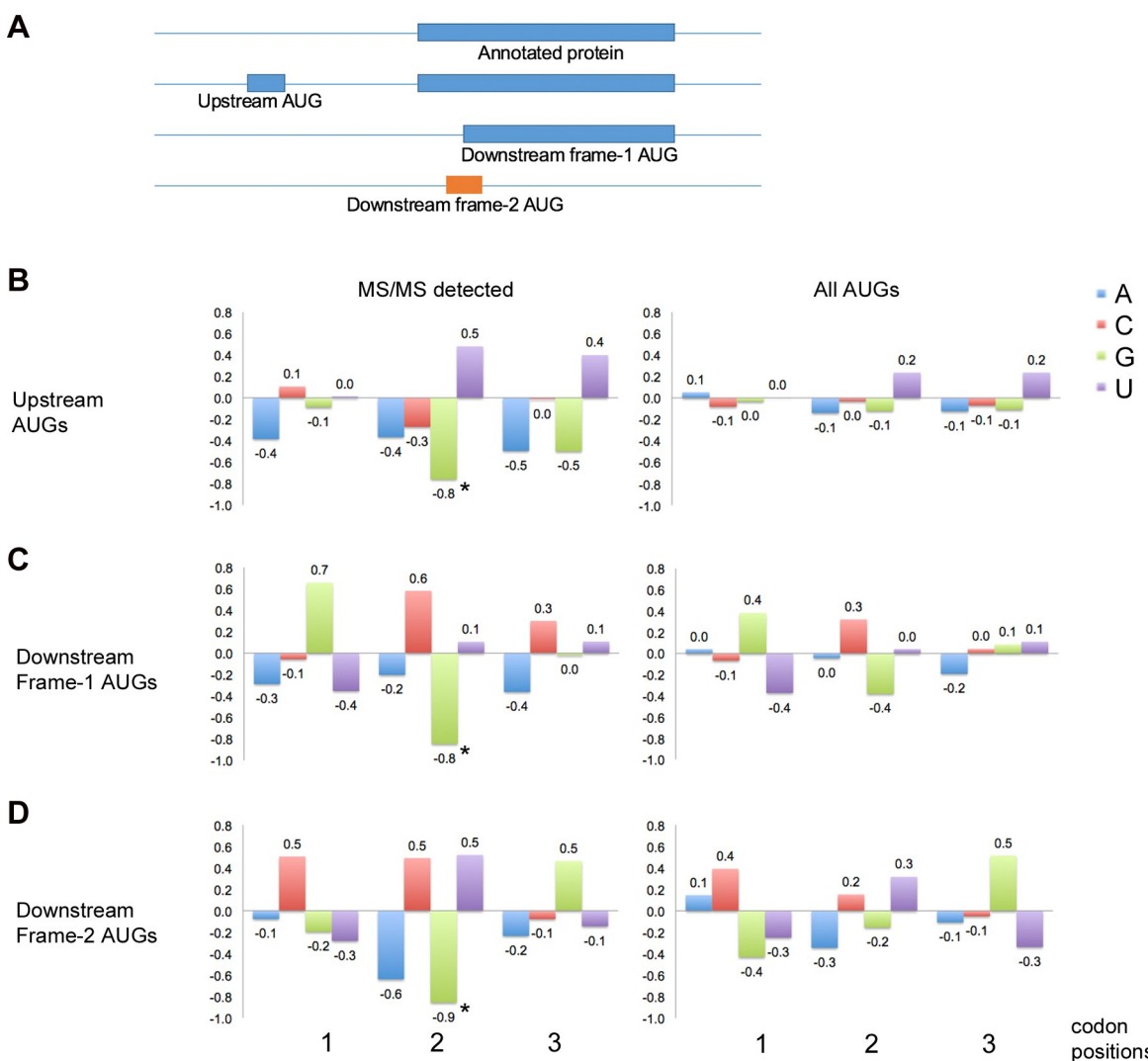

**Fig 3. GCN periodicity and non-standard translation sites.** (A) Amino-terminal peptides have been detected by MS/MS that map to translation initiation at AUG codons upstream and downstream of annotated start codons of genes (S1 File, S2 File) [32, 34]. (B, C and D) ORFs were aligned to the upstream AUGs (B; 34 genes) and downstream AUGs in frame 1 (C; 232 genes) or frame 2 (D; 248 genes) relative to the annotated frame-1 start codon. The average deviations from background, measured as $\log_2(\text{freq}_{obs}/\text{freq}_{background})$, are illustrated for codon positions 1, 2 and 3 for windows downstream of the start codon (positions 7 to 22). The 3-nucleotide periodicity is more pronounced for upstream AUGs that map to MS/MS-detected peptides compared to the full set of AUG triplets in known 5'UTRs [36, 37] (B). The periodicity is also more pronounced in-frame with downstream AUGs detected by MS/MS when compared to all frame-1 (C) and frame-2 (D) AUGs between nucleotides 2 and 100 after the annotated start codons (the full sets of AUGs screened in our MS/MS analysis; [32]). In particular, downstream of the frame-2 AUGs shows pronounced G2 depression (*) corresponding to the 3rd wobble position of frame 1. The G2 depression (* in B, C, and D) is significant by bootstrap analysis of randomly selected AUGs (from 2480 upstream AUGs (B), 2859 frame-1 downstream AUGs (C), and 4454 frame-2 downstream AUGs (D)). This suggests that G2 depression contributes to ribosome choice of start sites at upstream and downstream AUGs.

the same cell lysate samples to normalize immunoblot signals (Fig 4; S2 Fig). CRISPR-Cas9 gene editing was used to mutate the ramp regions of the two genes (see Methods). The ramp mutations were designed to increase or decrease the frequencies of G1 and C2 in codons 3 through 8 (Fig 4A) with the aim of making conservative amino acid substitutions as discussed below.

**Protein expression.** Ramp mutants with enhanced GCN periodicity were made by introducing point mutations into the ramp codons (SKN7::GCNpm, HMT1::GCNpm, Fig 4A) or

by insertion of four GCN-rich codons between codons 3 and 4 (SKN7::GCNi). Compared to the WT sequences, the three ramp mutants had between 4 and 6 extra G1 or C2 nucleotides in codons 3 through 8 (Fig 4A). All three mutants showed significantly reduced protein expression compared to their WT counterparts (Fig 4B and 4C). Multiple tests of the constructs confirmed that the depressed protein expression was statistically significant. Our observation of depressed protein levels was a little unexpected given that pronounced ramp GCN periodicity is associated with high-expression genes. However, this trend was supported further by our testing (described below) of ramp mutants with depressed periodicity.

Ramp mutants with depressed GCN periodicity were designed in three ways. (i) Through nucleotide substitutions, G was placed at position 2 of successive codons in the ramp region (SKN7::G2, HMT1::G2, Fig 4A) in contrast to the selection against G2 observed in the ramp regions of high-expression genes (Fig 2C and 2D); (ii) Also through nucleotide substitutions, an A-rich ramp was made (SKN7::A-rich) that had no G1 or C2 across the ramp codons (codons 3 through 9); (iii) Finally, the GCN repeat was shifted one nucleotide 3' across codons 3 through 6, giving a CNG repeat (HMT1::C1). These ramp mutants had between 2 and 4 fewer G1 or C2 nucleotides in codons 3 through 8 compared to WT (Fig 4A).

Depression of GCN periodicity led to increased protein expression. Upon insertion of G's at position 2 of codons (SKN7::G2, HMT1::G2), we observe a statistically significant increase in protein expression relative to SKN7 ($p<0.05$) and HMT1 (t test $p<0.001$) (Fig 4B and 4C). This is in striking contrast to the paucity of G at position 2 in highly-expressed genes. The A-rich mutant (SKN7::A-rich) also had significantly enhanced protein expression ($p<0.05$), and the CNG shifted mutant (HMT1::C1) also had slightly elevated protein levels although these increases were not significant.

The decreases and increases in protein expression observed when GCN periodicity was enhanced or depressed respectively, suggest that protein translation levels were correspondingly decreased and increased. However, in order to have confidence in these interpretations, it was important to assess whether the ramp mutations caused changes in mRNA levels, protein stability, or other properties, that might instead account for the observed changes in protein expression.

**mRNA levels.**   We examined mRNA levels in our GCN ramp mutants using RT-qPCR. We amplified TAP sequences at the 3' ends of the test mRNAs to measure their expression levels which were standardized relative to -dCT values of the *ALG9* mRNA which was used as a control [39]. The ramp mutants of *SKN7* and *HMT1* had similar mRNA expression compared to the WT genes (Fig 5A and 5B). Small changes in mRNA levels were detected that tended to be in the opposite direction to the changes in protein expression, suggesting the changes might be compensatory. HMT1::G2 and HMT1::C1 showed depressed mRNA levels (t-test $p < 0.05$). This contrasts to the increase in protein level of HMT1::G2. In summary, this RT-qPCR analysis indicated that the protein expression changes observed in our GCN ramp mutants are not explained by corresponding changes in mRNA levels.

**Codon and anticodon changes.**   The *SKN7* and *HMT1* ramp mutants have alterations in three or four codons that underlie the changes in protein expression (Fig 4). We designed the ramp mutants to have conservative amino acid substitutions where possible, as indicated by the modest Blosum scores [40] of the amino acid substitutions (S4 Table). In addition, comparisons of the normalized Translation Efficiency (nTE; [12]) scores at codons 3 through 8 (S3 Table) showed small differences in nTE scores compared to WT. The nTE score integrates codon preferences (cAI score; [41]) and tRNA availability (tAI score; [42]) capturing both the supply and demand for specific tRNAs. The insertion of four codons in *SKN7*::*GCNi* appeared to have more significant effects on the protein structures and its stability as described below.

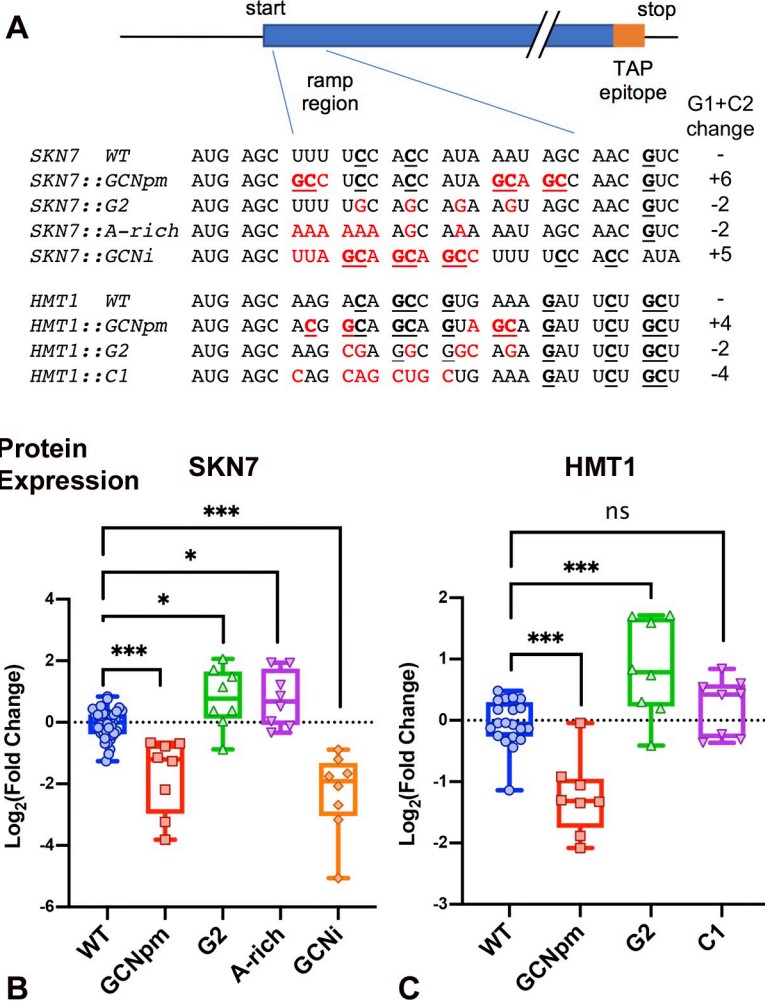

**Fig 4. Protein expression in ramp GCN mutants.** (A) Mutations were designed to increase or decrease GCN periodicity in the initial codons of test genes *SKN7* and *HMT1*. Nucleotides conforming to G and C at positions 1 and 2 of codons (G1 and C2) are underlined. The right column indicates the increase or decrease compared to WT in the frequency of G1 and C2 in codons 3 through 8. (B, C) Normalized densitometric quantification of Western blots testing protein expression in ramp mutants of *SKN7* (B) and *HMT1* (C). Each point represents a single biological replicate measure of the ratio of TAP signal to Tub1p signal, divided by the average of the WT ratio. Plotted is log₂(fold change) relative to the WT average. Boxes show 25 and 75% quartiles. T-test confidence levels in Figs 4 and 5: $p < 0.05^*$, $p < 0.01^{**}$ and $p < 0.001^{***}$.

The altered protein expression in the ramp mutants is associated with tRNA anticodon replacements that frequently change the identities and modifications of the wobble anticodon nucleotide (tRNA nucleotide 34; S5 Table). It is not yet known whether or not these changes affect the CAR-mRNA interactions, for example through alterations in anchoring of CAR to nucleotide 34, which may relate to the observation that the ramp mutants with depressed protein levels have more inosine 34 anticodons compared to WT, and mutants with elevated protein have fewer inosine 34 anticodons.

**Protein stability.** It was important to examine the stabilities of the proteins expressed from the ramp mutant constructs compared to WT. We examined protein stability of the ramp mutants by blocking protein synthesis with CYC and measuring the drop in protein level at 3 hours after the block (Fig 5C and 5D, S3 Fig). In general, protein stabilities of the

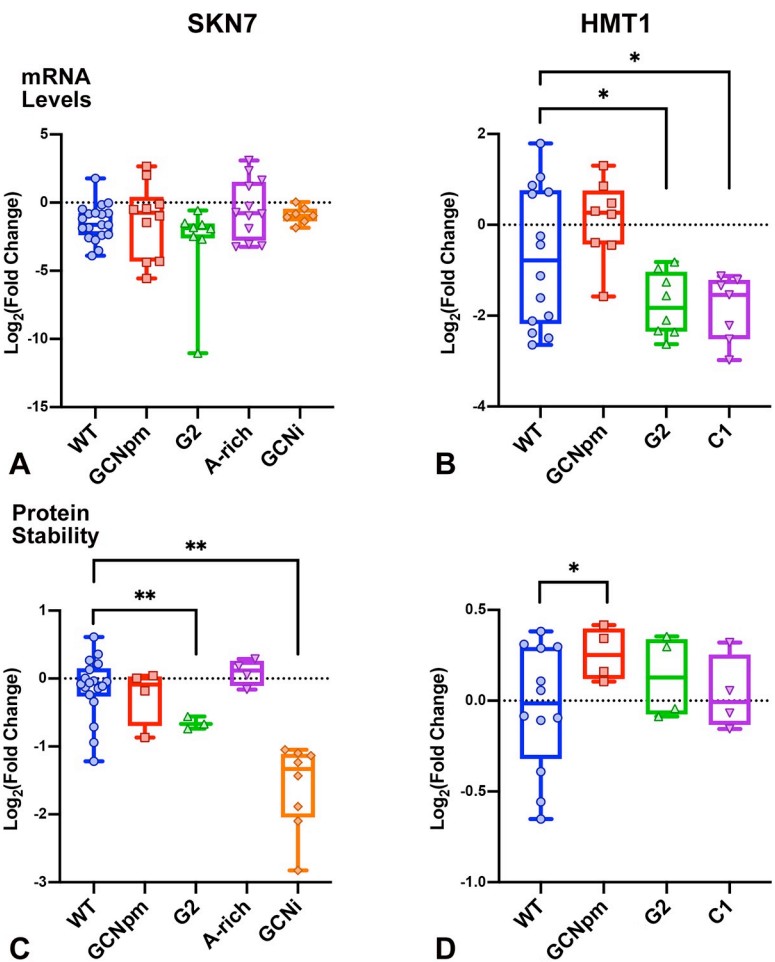

**Fig 5. Ramp mutant mRNA expression and protein stability.** (A, B) Box and whisker plots showing $-dC_T$ measurements for each mutant of *SKN7* (A) and *HTM1* (B). Each dot represents a single biological replicate averaged over technical duplicates. $-dC_T$ is proportional to the relative abundance of TAP sequence in cDNA pools prior to qPCR amplification plotted relative to $-dC_T$ for reference gene *ALG9*. Pairwise t-tests comparing each mutants' $-dC_T$ values to those obtained for WT determined that changes in mRNA abundance were not statistically significant except that HMT1::G2 and HMT1::C1 showed depressed mRNA levels ($p < 0.05^*$). (C, D) Protein stability levels were measured 3h after blocking protein synthesis. The proportion of protein present at 3h was calculated relative to the average 0h time point for WT or the appropriate mutant. These proportions were divided by the average proportion for WT to obtain normalized stabilities. Graphs show $\log_2$ of fold differences in protein stability compared to WT. The fold differences were not significant by t-test except for SKN7::G2 and SKN7::GCNi which had significantly lower stability ($p < 0.01^{**}$), and HTM1::GCNpm which was slightly stabilized ($p < 0.05^*$).

ramp mutants were similar to the WT proteins. However, SKN7::GCNi protein had significantly reduced stability (t-test $p < 0.01$, Fig 5C), perhaps a consequence of the 4-amino-acid insertion near the N-terminus, suggesting that the depressed protein levels of SKN7::GCNi without CYC (Fig 4B) may be largely explained by destabilization of the protein rather than reduced translation efficiency. Although SKN7::G2 may be a little less stable than WT ($p < 0.01$), the elevated protein levels without CYC (Fig 4B), a change in the opposite direction, suggests that the SKN7::G2 ramp mutation increases protein translation compared to WT. Moreover, while HMT1::GCNpm may be slightly stabilized ($p < 0.05$), the depressed protein levels without CYC (Fig 4B and 4C), again a change in the opposite direction, suggests that the HMT1::GCNpm ramp mutant depresses protein translation.

**mRNA secondary structure prediction.**   Computationally-predicted mRNA secondary structure surrounding the start codon has been shown to be associated with modulating translation [5]. Therefore, assessment of whether changes in protein expression might be explained by differences in predicted mRNA secondary structure between mutant and WT sequences was carried out using the m-FOLD web server [43]. Predicted secondary structures of nucleotides -10 to +60 relative to the AUG start codon were calculated using default parameters. The energies for the predicted structures are presented in S4 Fig for mutant and WT *SKN7* and *HMT1* mRNAs. It seems unlikely that translation levels would be significantly affected by the modest levels of predicted mRNA secondary structure.

**Translation levels inversely correlate with ramp GCN periodicity.**   Integrating our protein expression (Fig 4), mRNA expression (Fig 5A and 5B), and protein stability tests (Fig 5C and 5D), we conclude that all but one of the changes in protein expression detected in our GCN ramp mutants reflect corresponding changes in protein translation levels. Protein levels were depressed in ramp mutants with accentuated GCN—SKN7::GCNpm, SKN7::GCNi, HMT1::GCNpm (Fig 4B and 4C). The depression of SKN7::GCNi is probably largely due to destabilization of the expressed protein associated with the 4-amino-acid insertion (Fig 5C).

However, the depressions of SKN7::GCNpm and HMTI::GCNpm are not explained by protein destabilization (Fig 5C and 5D) nor altered mRNA levels (Fig 5A and 5B), and are most likely explained by depressed protein translation suggesting that the presence of G at position 1 and C at position 2 of codons in the ramp region tends to depress protein translation.

Supporting this interpretation, the four ramp mutants with reduced G1 and C2—SKN7::G2, SKN7::A-rich, HMT1::G2 and HMT1::C1— have increased protein levels (statistically significant for three of the four) that most likely reflect enhanced levels of protein translation. These constructs encode proteins with similar stabilities to their WT counterparts, and their mRNA expression levels are similar or lower than WT, suggesting that the higher detected protein levels reflect more efficient translation.

## G1 and C2 effects are context dependent

Our assessments of *SKN7* and *HMT1* ramp mutants in yeast cells suggested that enrichment of G1 and C2 in the ramp region codons tends to be associated with reduced protein translation levels. We compared this result with recent large-scale analyses of ramp sequence variants in *E. coli* [31] and *S. cerevisiae* [13]. Verma *et al.* [31] tested 215,414 different combinations of three (non-stop) codons, inserted at codon positions 3, 4 and 5 in the ramp region of a GFP reporter construct. Bacterial cells expressing these reporters were sorted into five bins based on reporter expression levels (GFP fluorescence). The distribution of each ramp sequence across the five bins allowed each nine-mer to be assigned a weighted expression level (on a continuous scale between 1 and 5). We partitioned the 215,414 nine-mers into 7 groups based on how many G1 and C2 nucleotides were in the three codons and examined the expression histogram for each group (Fig 6). The shapes of the expression curves suggest that at lower expression levels, higher densities G1 and C2 are beneficial for expression, whereas at higher expression levels, G1 and C2 are detrimental to expression. The latter trend is similar to our results with *SKN7* and *HMT1*. These opposite behaviors, depending on the expression range considered, provide the basis for a model for the function of the CAR interface discussed below (Fig 7).

Equivalent analysis of 34,914 nine-mer sequences inserted at codons 6–8 of a GFP reporter construct expressed in yeast cells [13] similarly showed that at low and intermediate protein levels, higher densities of G1 and C2 are beneficial for reporter expression (S5A Fig). However, the data lacked sufficient resolution to determine whether G1 and C2 are detrimental at higher expression levels since over 80% of the tested nine-mers were assigned to the top of three expression bins.

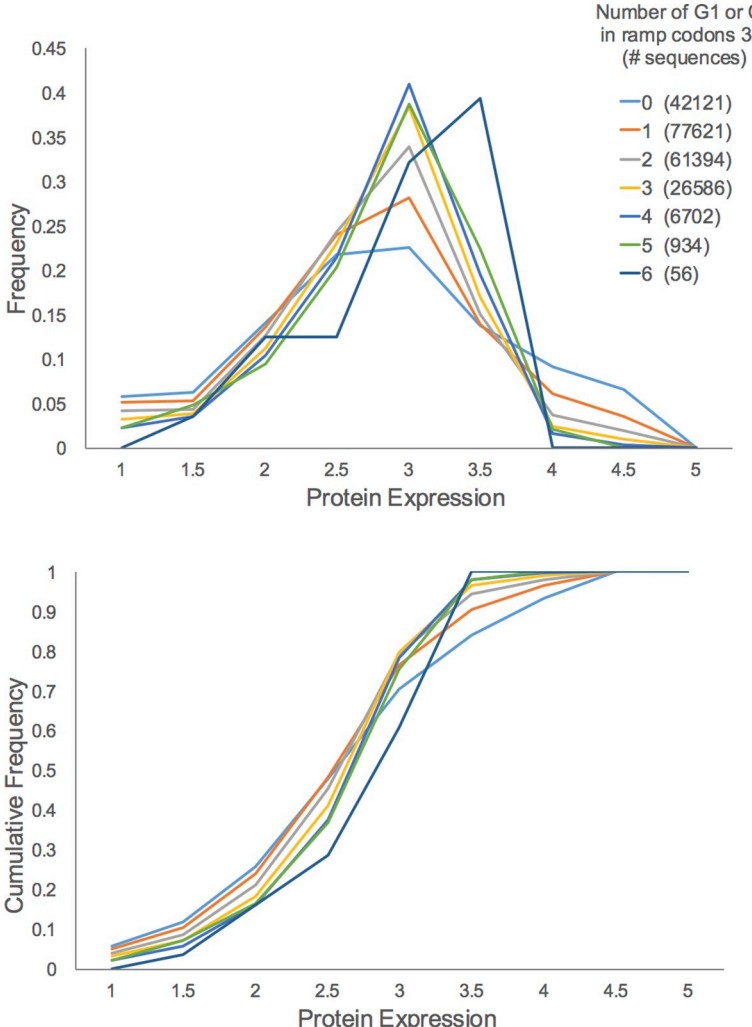

**Fig 6. G1+C2 assessment in ramp reporter genes.** Bacterial expression of 215,414 reporter constructs with different nine-mer sequences at codons 3–5 [31] allowed assignment of ramp sequences into nine expression ranges (1: 1< = x<1.5; 1.5: 1.5< = x<2;. . . 4.5: 4.5< = x<5; 5: x = 5). The nine-mers were divided into 7 groups based on the number of G1 and C2 nucleotides in the nine-mer, and the expression histogram of frequencies and cumulative frequencies for each group were graphed. The shapes of the graphs suggest that at lower levels of expression (~1 to 3), higher densities of G1 and C2 are beneficial to expression, but at higher expression levels (~3.5 to 5), G1 and C2 are detrimental.

Further assessment of both reporter data sets (S5 and S6 Figs) suggested that when pairs of ramp codons were selected for analysis, the effects of increasing G1+C2 density were more pronounced when the two codons were adjacent. For example, the effects were more pronounced for codons 6 and 7, or codons 7 and 8, compared to codons 6 and 8 (yeast experiment, S5B to S5D Fig). Similarly, the effects for codons 3 and 4, or codons 4 and 5, were more pronounced than for codons 3 and 5 (bacteria experiment, S6A to S6C Fig).

## Discussion

### Protein translation is sensitive to ramp GCN periodicity

We have explored the GCN periodicity observed in protein-coding ORFs, but not UTRs, of mRNA sequences. The GCN periodicity is pronounced in codons in the ramp region located

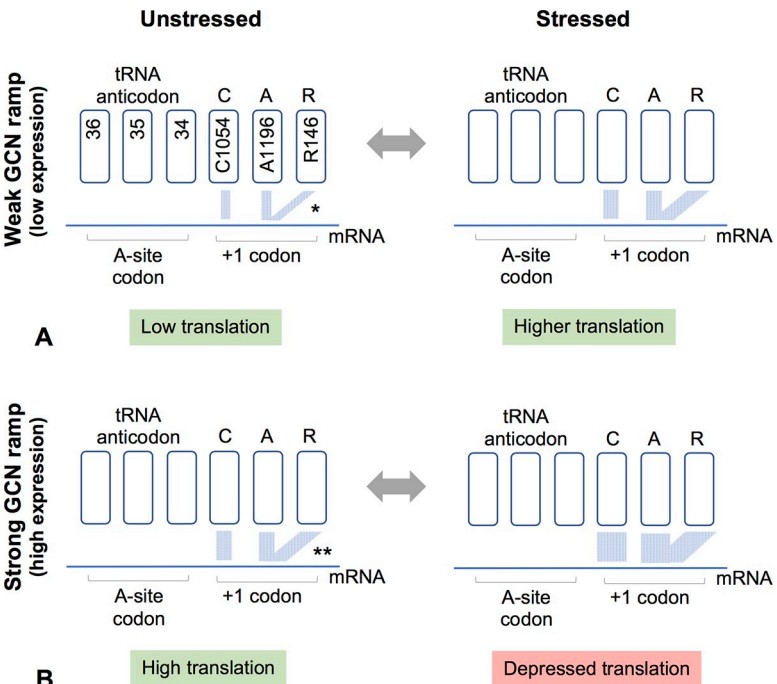

**Fig 7. A model for tuning translation efficiency.** Gene ramps with weaker GCN periodicity will tend to have weaker mRNA-CAR H-bond interactions (\*) whereas those with stronger periodicity will have stronger interactions (\*\*). We hypothesize that the strength of the mRNA-CAR interaction also depends on the modification states of the wobble anticodon nucleotide (tRNA nucleotide 34) and Rps3 R146, both of which are modulated by stress conditions. For example, our model assumes that in unstressed conditions (left), R146 is methylated and tends not to stack effectively with A1196, leading to weaker mRNA-CAR interactions; in stressed conditions (right), newly-made ribosomes have unmethylated R146 which stacks more consistently with A1196 giving strong mRNA-CAR interactions. When low-expression genes have modest GCN periodicity (A), overall mRNA-CAR interactions are relatively weak, and increasing those interactions stimulates translation; however, high-expression genes with more pronounced GCN periodicity (B) have strong mRNA-CAR interactions and show depressed translation when those interactions are increased. The enriched GCN ramp and consequent depressed translation of SKN7::GCNpm and HMT1:GCNpm mimics the effect of stress.

after translation start sites, particularly in highly-expressed genes (Fig 2), and is also accentuated after non-standard start sites (Fig 3). These observations suggest a functional relationship between GCN periodicity and protein translation. Yet when the ramp sequences were edited in two test genes—*SKN7* and *HMT1*— in their normal genomic locations, we found that enhancing ramp GCN periodicity leads to depressed translation, and reducing GCN periodicity has the opposite effect (Fig 4). While it is possible that *SKN7* and *HMT1* are not representative genes, these observations instead suggest a general model where GCN periodicity in the ramp region contributes to regulation of translation.

We hypothesize that (i) the ramp GCN periodicity helps ribosomes translate from non-standard start sites, perhaps by slowing down mRNA scanning allowing stabilization of translation; (ii) the ramp GCN periodicity marks high-expression genes for modulation of translation under different cell conditions. For example, a GCN-sensitive mechanism could mediate down-regulation of translation under stress conditions. Dampening of protein expression may be particularly beneficial for genes that otherwise have very high expression, potentially accounting for the correlation of pronounced ramp GCN periodicity with high-expression genes.

## The CAR surface may mediate GCN sensitivity

As introduced earlier, we have identified [27] a ribosome interaction surface located next to the decoding center A site that may mediate sensitivity to the ramp GCN periodicity (Fig 1). At the beginning of translocation, the C1054-A1196-R146 (CAR) interaction surface is anchored through base stacking to the wobble anticodon nucleotide (nucleotide 34) of the A-site tRNA, aligning the CAR surface so that it can H-bond with the +1 codon about to enter the A site. Our Molecular Dynamics (MD) experiments show that the CAR-mRNA interaction is weakened if GCN in the +1 codon is replaced with GGN, GAN or GUN.

We hypothesize that the CAR-mRNA interaction is regulated through modifications of Rps3 R146 or tRNA nucleotide 34 and that genes with ramp GCN periodicity are sensitive to this regulation. The guanidinium group of ribosomal protein S3 R146 is methylated by a methyl transferase, SFM1 [44, 45], that is down-regulated under stress conditions [46], suggesting that the CAR surface in ribosomes made during stress may have different mRNA interaction properties. Stress-dependent post transcriptional modifications of tRNA nucleotide 34 [47] may affect anchoring of the CAR surface to the tRNA and hence also affect CAR-mRNA interactions with the +1 codon.

## Tuned enhancement and depression of translation efficiency

As discussed earlier, many factors determine whether genes have high or low protein expression. We postulate that the CAR interaction with the +1 codon provides an additional layer of regulation that helps cells respond to different cell states.

Our assessment (Fig 6, S5 Fig) of a large-scale reporter gene analyses of ramp sequence variants expressed in bacteria [31] suggests that in the context of lower protein expression, enhanced G1+C2 nucleotide density in the ramp region is associated with elevated protein expression (see Fig 6 legend), whereas in the context of high protein expression, increased G1 +C2 density is associated with depressed protein levels, as observed in our analysis of ramp mutants of *SKN7* and *HMT1*. This suggests that the ribosome response to increasing the strength of the mRNA-CAR interaction switches between activation (an accelerator) and inhibition (a braking system) depending on whether translation levels are low or high. An attractive extension of this idea is that the strength of the mRNA-CAR interaction, and hence the strength of the accelerator or braking system, can be modulated through R146 methylation or modification of tRNA nucleotide 34 in response to stress conditions. This model is presented in Fig 7. Since prokaryotes appear not to have the equivalent of R146 in ribosomal protein S3, if they have a similar interaction surface, it might consist of C1054 and A1196, and be modulated through tRNA nucleotide 34 modifications. While molecular details remain to be determined, our study suggests that GCN-sensitive interactions between the CAR interface and the +1 codon of the mRNA contributes to the cell's ability to tune translation rates in response to cellular conditions.

## Methods

### ORF sequence analysis

Genomic sequences of *S. cerevisiae* genes (verified and unverified, https://www.yeastgenome. org) with introns removed were imported into a relational database and SQL scripts were used to align sequences at their annotated start or stop codons, or AUG triplets upstream or downstream of the annotated start. Frequencies and Log-Odds Ratios (LOR) of nucleotides at each alignment position were computed using an information calculator (http://igs.wesleyan.edu)

or Python scripts. The frequencies of nucleotides in all annotated ORFs (A: 0.326; C: 0.192; G: 0.204; T: 0.278) were used as background frequencies for LOR calculations.

## Ramp mutant construction

Ramp mutants of *SKN7* and *HMT1* were made using CRISPR-Cas9 gene editing [48] in a modified C-terminal TAP-epitope-tagged line of *SKN7* or *HMT1* [33] (in which the *His3MX6* selectable marker was replaced with *URA3*). A *KanMX6* cassette was inserted by homologous recombination in place of nucleotides 8 to 22 after the start codon. The *KanMX6* cassette was then targeted for cleavage by CRISPR-Cas9 [48] expressed from a transformed plasmid (*pRS425-Cas9-hphMX*) in the presence of a healing fragment containing the desired mutated sequences (S1 and S2 Tables).

## Western analysis

Protein expression levels were determined in Western analysis performed with extracts of dividing cells (after 1 day of culture) as described [32, 34]. At least four independent cultures for each ramp mutant were inoculated in 10mL YPD liquid media alongside four WT *SKN7-TAP-URA3* or *HMT1-TAP-URA3*. After 18–24 hours of growth at 30°C, 5mL of cells were isolated and washed with 400µL and then 200µL of 20% trichloroacetic acid (TCA) and lysed by vortexing with glass beads. After precipitation with TCA, protein pellets were washed twice with acetone and dried overnight before being weighed.

Total protein extracts were then prepared for separation by SDS-PAGE. Laemmli sample buffer (2X) was added to protein pellets to obtain a total protein concentration of 50mg/mL. 1µL of 4M NaOH was added to each sample to neutralize pH. Samples were then vortexed for 15–30 seconds, boiled to denature proteins, re-vortexed and centrifuged at 13,200rpm to remove cell debris. Total supernatant protein concentrations were measured (absorbance at 260nm) using a Nanodrop 4000. Total protein (200ug) was loaded into wells of 10% Tris-Glycine polyacrylamide gels (Invitrogen WedgeWell XP00102BOX) and electrophoresed at 4°C at 100V. Protein samples were transferred onto PVDF membranes (Invitrogen Novex IB24001) using an iBlot2.0 fast transfer machine at 18V for 7 minutes. Membranes were blocked overnight with 10mL of 5% milk protein in Tris-Buffered Saline-Tween (TBS-T; 20mM Tris, 150mM NaCl, 0.5% Tween-20) at 4°C. After blocking, membranes were washed 3 X 5 minutes in 1X TBS-T by shaking (110rpm) at room temperature. Membranes were then probed for 1 hour with 1.5µL rabbit anti-TAP polyclonal antibody (ThermoFisher CAB1001) diluted in 7.5mL of 1% milk in TBS-T (1:5000 dilution). After three washes in 1X TBS-T, secondary antibody was added (Thermo 31460, Goat anti-rabbit IgG (H+L) HRP polyclonal antibody) for 1 hr followed by 3 washes with TBS-T.

Blots were visualized using the GE Amersham ECL Prime Western Blotting Detection Reagent (Fisher Scientific) and imaged in a Syngene G:Box. SynGene software was used to verify that signals did not exceed the linear dynamic range of detection. Pixel density was quantified for each sample using ImageJ. Prior to quantitation, images were converted to a 16-bit format and background signal was subtracted using a sliding-paraboloid algorithm [49]. Pixel density was plotted for each lane and the area under the curve was calculated as a proxy measurement for protein abundance.

As a loading control, membranes were probed for Tub1. Before re-probing, membranes were stripped 2X10 min with a mild stripping buffer (pH 2.2; 0.2 M glycine, 0.5% Tween-20, and 3 mM sodium dodecylsulfate), washed 2X 10-min in 1X phosphate-buffered saline (PBS; 137 mM NaCl, 2.7 mM KCl, 10 mM $Na_2HPO_4$ and 1.8 mM $KH_2PO_4$), and washed 2x 5 mins with TBS-T, before blocking in 5% milk overnight at 4°C. Membranes were probed with anti-

Tubulin rabbit monoclonal antibody (1:5000 dilution, Abcam ab184966) and the same secondary antibody as above. In order to scale analyses across multiple replicates, TAP:Tub1 signal ratios were divided by the average ratio of WT samples allowing comparison of ramp mutant protein expression relative to the WT average across all experiments. At least eight biological replicates were tested for each ramp mutant genotype. Normalized ratios were plotted using Prism and analyzed using a student's T-test.

## Quantitative RT-PCR

The levels of mRNA expression in cell extracts were determined using quantitative RT-PCR. For all RT-qPCR analyses, two 1.7mL yeast cell samples were isolated in parallel with Western blot samples. These samples were centrifuged for 1 minute at 13,200rpm, supernatant discarded, and cell pellets were flash-frozen in liquid nitrogen. Total RNA was isolated from pellets with a hot acidic phenol extraction method. Pellets were thawed on ice and resuspended in 400μL of acidic phenol (pH 4.5) and 400μL of TES solution (10 mM Tris-HCl pH 7.5, 10mM EDTA, 0.5% SDS in DEPC-treated water). The mixture was vortexed for 10 seconds, incubated at 65˚C for 1 hour, placed on ice for 5 minutes, and then centrifuged at 14,000rpm at 4˚C for 5 minutes. The aqueous phase was saved, 400μL of acidic phenol was added, and the centrifugation steps were repeated. The aqueous phase of this second phenol wash was mixed with 400μL of chloroform and re-centrifuged at 4˚C for 5 minutes. The aqueous layer was then mixed with 3M sodium acetate and ice-cold ethanol to precipitate nucleic acids. Pellets were dissolved in 200μL of DEPC-treated water. Concentrations were measured using a Nanodrop 4000. 1 ug of total nucleic acid per sample was run on a 1.6% agarose gel to assess the quality of total RNA extractions.

RNA samples (2.5μg) were treated with DNAse I treatment (Ambion #AM1906) and the absence of genomic DNA was verified by PCR using forward and reverse primers which anneal within the TAP epitope tag's DNA sequence. cDNA copies of mRNA molecules were made by treating 0.5μg of DNAse-treated RNA with reverse transcriptase (Fisher #28025013). After 3 minutes heating at 85˚C, Oligo-dT primers were annealed to mRNA poly-A tails followed by a 60-minute extension reaction at 42˚C with 100units/μL reverse transcriptase, 10units/μL RNAse inhibitor, 1X RT buffer (250mM Tris-HCl (pH 8.3), 375mM KCl, 15mM $MgCl_2$), and 10mM dNTPs. After extension, samples were heated at 92˚C for 10 minutes and the presence of cDNA was tested by PCR.

Samples were subjected to a qPCR reaction using a SYBR Green amplification mix and detected on an Applied Biosystems StepOne Plus machine. PCR reactions were carried out with primers which anneal within the TAP sequence. The qPCR analyses were normalized to $C_T$ values obtained for the reference gene *ALG9* [39]. $C_T$ values for each sample were averaged across the technical duplicates and used to calculate $dC_T$, the difference in cycle threshold between the target TAP sequence and the reference *ALG9* sequence for each sample. These values were tested for statistical significance between WT and ramp mutant strains. Average fold change in mRNA abundance was calculated using the $2^{-ddCT}$ method [50].

## Assessment of protein stability

To assess whether ramp mutants of SKN7 or HMT1 had altered protein stability, cells were treated with cycloheximide (CYC; [51, 52]) to block protein synthesis and the reductions in protein levels measured after 3 and 6 hours of CYC treatment. Cultures of cells expressing the constructs of interest were grown overnight, and after diluting the cultures to an $OD_{600}$ of 0.2, cells were grown for 3 hours, and ~$3.3X10^8$ cells were brought up in 5 ml YPD with CYC (10 mg/ml). Cells were collected at 0, 3, and 6 hours after the addition of CYC and frozen in a dry

ice-ethanol bath. Protein expression levels for each time-point were measured using the same Western protocol described above with the following modifications in the protein extraction protocol. Instead of using glass beads, 2M NaOH, 1M β-mercaptoethanol solution was used to lyse cells. 50% TCA was added to precipitate the proteins, and the protein pellets were washed with acetone and dried overnight at 4˚C. Protein extracts were brought up to 100 μL in Laemmli sample buffer (2X) and 1 μL of 2M Tris-Base was added to neutralize any remaining TCA.

### Large-scale reporter gene analysis

Python scripts were used to partition expression of GFP reporter genes with different nine-mers at codons 3–5 (*E. coli* data, [31]) or codons 6–8 (*S. cerevisiae* data, [13]). The nine-mer sequences were partitioned according to reporter expression levels and the sum of the number of G1 and C2 nucleotides at positions 1 and 2 respectively of the 3 codons in the nine-mer. The yeast data excluded ramp sequences designated [13] to have potential secondary RNA structure.

## Supporting information

**S1 Fig. Global protein expression.** Histogram and density plot of the protein expression levels for all yeast genes, as measured by [33]. Protein expression levels for SKN7 and HMT1 are indicated.
(PDF)

**S2 Fig. Protein expression in ramp GCN mutants.** Western images used in quantitation described in Fig 4. These images have exposure times ranging from ten seconds to five minutes. The images within the linear dynamic range of detection with the least background signal were used in densitometric analyses. The images are separated by mutant strain. Each lane represents an independently-grown culture for that strain. TAP and Tub1 (α-tubulin) images are paired for each gel. Note that in some experiments, incomplete stripping after the first probing led to some retention of band signals. Also, with the anti-tubulin incubations, the TAP protein can also be visible because the secondary antibody has an IgG domain that recognizes the Protein A region of the TAP epitope tag; this was most pronounced for the more abundant HMT1 protein.
(PDF)

**S3 Fig. Protein stability in ramp GCN mutants.** To measure protein stability, protein synthesis was blocked with cycloheximide (CYC) and samples collected at 0, 3 and 6h. Protein levels after 3h were used to compare protein stability (see Fig 5). As a loading control for SKN7-TAP quantitation, samples were spiked with HTM1-TAP (that was not exposed to CYC). As a loading control for HTM1 quantitation, blots were probed for Tub1 (α-tubulin; Tub antibody) protein which has a half life of 20h. Blots were probed with anti-TUB and then anti-TAP. The ramp mutant proteins were each expressed in multiple independent cultures (# labels).
(PDF)

**S4 Fig. RNA secondary structure prediction.** Predicted RNA secondary structures for *SKN7* (A) and *HMT1* (B) mutants and wildtype sequence. The lowest energy structures predicted by mFOLD for each strain is represented by a hairpin diagram. The energy of this structure is reported in the fourth column. In the last column, nucleotides -10 to +70 relative to the AUG start codon are presented above dot-bracket representations of all secondary structures reported by mFOLD using default parameters. The dot-bracket representation is followed by the corresponding dG free energy value for that structure. The predicted secondary structures

for *HMT1::G2* and *HMT1::GCNpm* have slightly lower dG values than the other predicted structures.
(PDF)

**S5 Fig. Large-scale reporter analysis in yeast.** Yeast expression of 35,811 reporter constructs with different nine-mer sequences at codons 6–8 [13]. Reporter expression is divided into three bins, low, intermediate and high. Expression is graphed as described in S6 Fig. The shapes of the graphs suggest that at lower levels of expression, higher densities of G1 and C2 are beneficial. The effect is more pronounced for codons 6 and 7 (B) or codons 7 and 8 (C) compared to codons 6 and 8 (D) suggesting that codon adjacency may increase the effect. The data do not show whether with higher expression levels, G1 and C2 are detrimental to expression, because the high expression bin has over 80% of the reporter sequences and therefore has limited resolution.
(PDF)

**S6 Fig. Large-scale reporter analysis in bacteria.** Bacterial expression of 215,414 reporter constructs with different nine-mer sequences at codons 3–5 [31] allowed assignment of ramp sequences into nine expression ranges (1: $1< = x<1.5$; 1.5: $1.5< = x<2$;... 4.5: $4.5< = x<5$; 5: $x = 5$). The nine-mers were divided into 5 groups based on the number of G1 and C2 nucleotides in two codons: codons 3 and 4 (A), codons 4 and 5 (B) or codons 3 and 5 (C). The expression histogram (left) and cumulative frequencies (right) for each group were graphed. The shapes of the graphs suggest that at lower levels of expression, higher densities of G1 and C2 are beneficial, but at higher expression levels, G1 and C2 are detrimental to expression. These effects are slightly more pronounced for codons 3 and 4 (A), compared to codons 4 and 5 (B), suggesting that codon 3 may have a little more influence than codon 4. The effects are also slightly more pronounced for codons 3 and 4 (A) compared to codons 3 and 5 (C) suggesting that codon adjacency may increase the effect.
(PDF)

**S1 Table. Yeast strains for ramp mutants.** Yeast strains for ramp mutants. Strains used in this study were generated as described in Methods. The table includes the name, strain ID, colony number, and genotype for each yeast strain used in this study. The first thirty nucleotides of the candidate gene sequence is included for each strain, with deviations from wildtype sequence highlighted in red lettering.
(PDF)

**S2 Table. Primers for ramp mutants.**
(PDF)

**S3 Table. Normalized Translation Efficiency.** Normalized Translation Efficiency. Codon-specific normalized translation efficiency (nTE) scores for each mutant in SKN7 (A) and HMT1 (B). Green scores represent positive deviations from wildtype sequence, and red scores represent negative deviations from wildtype sequence. Normalized Translation Efficiency (nTE) scores provided a measure of codon bias (cAI) and tRNA availability (tAI).
(PDF)

**S4 Table. BLOSUM assessment.** BLOSUM scores for amino acid substitutions for point mutants of SKN7 and HMT1. Red letters represent amino acid deviations from wildtype protein sequence. BLOSUM62 scores are sums of each value obtained from the matrix for each amino acid substitution.
(PDF)

**S5 Table. Ramp mutant anti-codons.** Comparison of anticodons with modifications.
(PDF)

**S1 File. uORF spectra data.** MS/MS data collected as described [32].
(XLSX)

**S2 File. uORF MS/MS spectra.** Spectra for data in S1 File.
(PDF)

**S3 File. Western data for Fig 4.**
(XLSX)

**S4 File. qRT-PCR data for Fig 5.**
(XLSX)

**S5 File. Western data for Fig 5.**
(XLSX)

## Acknowledgments

We thank Joe Coolon, Amy MacQueen, Scott Holmes, Ruth Johnson, Carol Dalgarno, Abdel Elsayed, Clara Nachmanoff, and Audrey McMahon for discussions.

## Author Contributions

**Conceptualization:** William A. Barr, Ruchi B. Sheth, Jack Kwon, Jacob W. Glickman, Felix Hart, Om K. Chatterji, Kristen Scopino, Karen Voelkel-Meiman, Daniel Krizanc, Kelly M. Thayer, Michael P. Weir.

**Data curation:** William A. Barr, Ruchi B. Sheth, Jack Kwon, Jacob W. Glickman, Michael P. Weir.

**Formal analysis:** William A. Barr, Jack Kwon, Jacob W. Glickman, Kristen Scopino, Michael P. Weir.

**Funding acquisition:** Michael P. Weir.

**Investigation:** William A. Barr, Ruchi B. Sheth, Jack Kwon, Jungwoo Cho, Felix Hart, Om K. Chatterji, Kristen Scopino, Karen Voelkel-Meiman, Kelly M. Thayer, Michael P. Weir.

**Methodology:** William A. Barr, Ruchi B. Sheth, Jack Kwon, Jungwoo Cho, Jacob W. Glickman, Felix Hart, Om K. Chatterji, Kristen Scopino, Karen Voelkel-Meiman, Daniel Krizanc, Kelly M. Thayer, Michael P. Weir.

**Project administration:** Michael P. Weir.

**Supervision:** Michael P. Weir.

**Visualization:** Jacob W. Glickman, Michael P. Weir.

**Writing – original draft:** William A. Barr, Michael P. Weir.

**Writing – review & editing:** William A. Barr, Ruchi B. Sheth, Jack Kwon, Jungwoo Cho, Jacob W. Glickman, Felix Hart, Om K. Chatterji, Kristen Scopino, Karen Voelkel-Meiman, Daniel Krizanc, Kelly M. Thayer, Michael P. Weir.

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
