## [Decision Letter · Decision Letter 0]

21 Aug 2020

GCN sensitive protein translation in yeast

PONE-D-20-12392

Dear Dr. Weir,

We’re pleased to inform you that your manuscript has been judged scientifically suitable for publication and will be formally accepted for publication once it meets all outstanding technical requirements.

Kind regards,

Arthur J. Lustig

Academic Editor

PLOS ONE

Additional Editor Comments (optional):

Reviewers' comments:

Reviewer's Responses to Questions

**Comments to the Author**

1. Is the manuscript technically sound, and do the data support the conclusions?

Reviewer #1: Yes

Reviewer #2: Yes

2. Has the statistical analysis been performed appropriately and rigorously? 

Reviewer #1: Yes

Reviewer #2: Yes

3. Have the authors made all data underlying the findings in their manuscript fully available?

Reviewer #1: Yes

Reviewer #2: Yes

4. Is the manuscript presented in an intelligible fashion and written in standard English?

Reviewer #1: Yes

Reviewer #2: Yes

5. Review Comments to the Author

Reviewer #1: The paper by Barr et al. shows an elaborated example for research spanning various disciplines, bringing together molecular dynamics, bioinformatics and wet-lab biochemistry - each technically very thorough.

The conclusions drawn are well-based and elucidate a phenomenon so far little understood.

Reviewer #2: The manuscript by Barr et al. is devoted to the discovery and study of the mechanism of mRNA translation regulation, mediated by overrepresentation of GCN codons in the initial section of ORFs (so-called ramp region). The authors make a counter-intuitive observation that while GCN periodicity is pronounced in mRNA of genes with high expression, its de novo artificial introduction into the ramp sequence reduces the translation efficiency. Adequacy of the contribution of GCN codons representation to the translation level of the model genes’ (SKN7 and HMT1 and their sequence-altered variants) mRNA is proved by the scrupulous control experiments aimed at the transcription efficiency and protein stability measurements as well as the normalized translation efficiency scoring. The authors put forward a theory that translation levels inversely correlate with ramp GCN periodicity, confirming it not only with their model experiments on a pair of yeast genes, but also with the analysis of large-scale data previously published by other groups studying the role of the ramp region in translation efficiency. Finally, the authors propose a mechanistic model for the interaction of ribosome with mRNA (CAR-mRNA), which explains the regulatory principle of GCN periodicity (metaphorically described as the accelerator and braking system for ribosome translation).

It was a pleasure for me to read this manuscript. In form and in content significance, it recalls the classic papers of the 60-70s on the discovery of the basic principles of molecular biology. It is surprising that such fundamental and seemingly self-explanatory aspect of gene expression will be included in the coursebooks only in the third decade of the 21st century.

Undoubtedly, the theory proposed by the authors requires further testing of its applicability on the larger gene cohorts and within the translation systems of different organisms. Also these results call for the structural studies of the CAR-mRNA interface. However, to my mind the need for further development of this topic does not detract from the contribution already made by Barr et al. team. I would strongly recommend this manuscript for publication as it is.

6. PLOS authors have the option to publish the peer review history of their article (what does this mean?). If published, this will include your full peer review and any attached files.

Reviewer #1: **Yes: **Andreas Savelsbergh

Reviewer #2: No

---

## [Editor Report · Acceptance letter]

28 Aug 2020

PONE-D-20-12392 

GCN sensitive protein translation in yeast 

Dear Dr. Weir:

I'm pleased to inform you that your manuscript has been deemed suitable for publication in PLOS ONE. Congratulations! Your manuscript is now with our production department. 

Kind regards, 

on behalf of

Dr. Arthur J. Lustig 

Academic Editor

PLOS ONE